# The Relationship between Pathogenesis and Possible Treatments for the MASLD-Cirrhosis Spectrum

**DOI:** 10.3390/ijms25084397

**Published:** 2024-04-16

**Authors:** Paulina Vidal-Cevallos, Adriana P. Sorroza-Martínez, Norberto C. Chávez-Tapia, Misael Uribe, Eduardo E. Montalvo-Javé, Natalia Nuño-Lámbarri

**Affiliations:** 1Obesity and Digestive Diseases Unit, Medica Sur Clinic & Foundation, Mexico City 14050, Mexico; paulina.vice@gmail.com (P.V.-C.); nchavezt@medicasur.org.mx (N.C.C.-T.); muribe@medicasur.org.mx (M.U.); montalvoeduardo@hotmail.com (E.E.M.-J.); 2Translational Research Unit, Medica Sur Clinic & Foundation, Mexico City 14050, Mexico; paola.sorrozamtz@gmail.com; 3Department of Surgery, Faculty of Medicine, Universidad Nacional Autónoma de Mexico, Mexico City 04360, Mexico; 4Hepatopancreatobiliary Clinic, Department of Surgery, Hospital General de Mexico “Dr. Eduardo Liceaga”, Mexico City 06720, Mexico

**Keywords:** metabolic syndrome, sedentarism, fibrosis, hepatic steatosis, steatohepatitis, diet

## Abstract

Metabolic dysfunction-associated steatotic liver disease (MASLD) is a term that entails a broad spectrum of conditions that vary in severity. Its development is influenced by multiple factors such as environment, microbiome, comorbidities, and genetic factors. MASLD is closely related to metabolic syndrome as it is caused by an alteration in the metabolism of fatty acids due to the accumulation of lipids because of an imbalance between its absorption and elimination in the liver. Its progression to fibrosis is due to a constant flow of fatty acids through the mitochondria and the inability of the liver to slow down this metabolic load, which generates oxidative stress and lipid peroxidation, triggering cell death. The development and progression of MASLD are closely related to unhealthy lifestyle habits, and nutritional epigenetic and genetic mechanisms have also been implicated. Currently, lifestyle modification is the first-line treatment for MASLD and nonalcoholic steatohepatitis; weight loss of ≥10% produces resolution of steatohepatitis and fibrosis regression. In many patients, body weight reduction cannot be achieved; therefore, pharmacological treatment should be offered in particular populations.

## 1. Introduction

Metabolic dysfunction-associated steatotic liver disease (MASLD) is the latest term for steatotic liver disease associated with metabolic syndrome [1]. There was consensus to change the definition to include the presence of at least one of five cardiometabolic risk factors. Those with no metabolic parameters and no known cause were deemed to have cryptogenic steatotic liver disease [2]. It is considered the most common chronic liver disease in the world and, nowadays, is considered the most common cause of liver transplantation in the world [3].

The phenomenon of hepatic steatosis originates from an alteration in the metabolism of the fatty acids used for the constitution of hepatic triacylglycerol (TAG) from three sources: diet, adipose tissue, and de novo synthesis [4]. Donnelly et al. reported that 15% of liver fat is derived from the diet, 26% from de novo lipogenesis, and 59% from circulating free fatty acids. The accumulation of lipids results from the imbalance between its uptake and removal pathways in the liver [5]. Excessive consumption of saturated fats and simple carbohydrates contributes to the formation of hepatic steatosis through complex metabolic pathways involving TAG synthesis and de novo lipogenesis [6]. Studies in humans indicate that TAG arises primarily from the diet, and up to 20% of fatty acids are secreted as very low-density lipoproteins (VLDL) within six hours of ingestion [7].

A study by the American Diabetes Association that compared the effect of different diets, one high in carbohydrates, another high in saturated fats, and the last high in unsaturated fats, showed that a diet rich in saturated fats and simple carbohydrates causes an increase of intrahepatic TAG derived from lipolysis, in contrast to the diet rich in unsaturated fats [8]. Excess lipolysis generates most of the fatty acids for intrahepatic TAG synthesis, promoted by a high-fat diet due to the high production of glycerol and free fatty acids.

Fructose is a highly lipogenic nutrient that increases TAG plasma levels, primarily after acute ingestion [9]. On the other hand, fructose can directly modulate nuclear receptors, regulating de novo lipogenesis and intervening in the development of MASLD [10].

Increased fatty acids induce overloaded b-oxidation and subsequent mitochondrial dysfunction due to the accumulation of lipotoxic intermediates [11]. In MASLD, there is a loss of mitochondrial cristae and dysregulation in fatty acid oxidation. A clinical study demonstrates that patients with MASLD have a reduction in palmitic acid oxidation compared to healthy subjects [12].

The inflammatory response in steatohepatitis is not entirely elucidated at the molecular level; however, it is known that it begins with endothelial and Kupffer cells (macrophages located in the liver), which are activated by different extrahepatic stimuli, one of them generated by toxic lipids [13]. Inflammasomes are secreted, which are intracellular pattern recognition receptors that serve as a sensor of the immune system, responsible for the maturation of inflammatory cytokines and chemokines, such as interleukin-1B, interleukin-6, interleukin-18, and the tumor necrosis factor [14], leading to oxidative stress, endoplasmic reticulum stress, and mitochondrial dysfunction [15].

## 2. Risk Factors

### 2.1. Obesity and Insulin Resistance

Obesity stands as the primary contributor to MASLD, with a profound interplay between adipose tissue and endocrine function impacting metabolic balance [8]. For instance, low levels of adiponectin, a key regulator of peroxisome proliferator-activated receptor gamma (PPARγ) highly expressed in adipocytes, are linked with various components of metabolic syndrome, including visceral adipose tissue accumulation, hyperlipidemia, insulin resistance, and type 2 diabetes [16,17,18].

PPARγ plays a critical role in liver function, particularly in regulating glucose and lipid metabolism by inducing proteins involved in fatty acid uptake, binding, and transport. Its activity promotes intrahepatic lipid accumulation by upregulating lipogenic gene expression, notably SREBP-1c (sterol regulatory element binding protein), thereby influencing liver physiology [19]. Insulin resistance further exacerbates hepatic steatosis, as insulin’s ability to suppress adipose tissue lipolysis diminishes while simultaneously activating de novo lipogenesis pathways through transcription factors like SREBP-1c and carbohydrate-responsive element binding protein (ChREBP) [20]. Glucose also plays a role in stimulating de novo lipogenesis, initiating the expression of glycolytic and lipogenic genes necessary for fatty acid synthesis [21].

Individuals with steatohepatitis often exhibit reduced insulin sensitivity and heightened insulin secretion [22]. Additionally, diabetes mellitus exacerbates fibrosis progression [23]. Studies indicate a significantly elevated risk of incident type 2 diabetes mellitus and metabolic syndrome in MASLD patients [24].

Furthermore, MASLD correlates with adverse cardiovascular outcomes, including increased carotid intima-media thickness, coronary calcification, and endothelial dysfunction, which contribute to arterial stiffness [25]. Meta-analyses have underscored MASLD’s association with heightened risk for fatal and nonfatal cardiovascular events, alongside altered levels of triglycerides, high-density lipoprotein (HDL), and adiponectin [26,27].

These insights highlight the intricate relationship between obesity, insulin resistance, and MASLD, emphasizing the need for comprehensive management strategies addressing both metabolic and cardiovascular risk factors.

### 2.2. Sedentarism

Sedentary behavior encompasses activities characterized by a sitting or reclining posture with minimal energy expenditure. Numerous studies have underscored the adverse links between prolonged sedentary time and metabolic health outcomes, notably contributing to excessive weight gain and liver disorders [28]. The development of MASLD is intimately intertwined with unhealthy lifestyle choices, including poor dietary habits and physical inactivity [29].

Extended periods of sedentary behavior have been correlated with compromised metabolic health and increased liver fat accumulation. Research suggests that being inactive for more than three hours daily heightens the susceptibility to developing metabolic syndrome [30]. Moreover, each additional hour of sedentary time has been associated with a heightened probability of metabolic syndrome and increased storage of abdominal, subcutaneous, and hepatic fat [29].

Conversely, individuals with MASLD exhibit low levels of physical activity [28]. Aerobic deconditioning is prevalent among subjects with steatohepatitis, indicating that activity levels in these patients often fall below recommended physical fitness guidelines [31].

### 2.3. Alcohol

The metabolic-alcohol-related liver disease (MetALD) is a new term to classify patients who exhibit both MASLD and consume a significant amount of alcohol. MetALD is defined as individuals who consume more than 140 to 350 g of alcohol per week for females and more than 210 to 420 g per week for males. This classification emphasizes the importance of alcohol as a contributor to liver disease pathogenesis. By defining MetALD, the need for further research is highlighted to understand this patient population’s characteristics and outcomes better [32].

In a cross-sectional analysis involving 40,189 patients from the UK Biobank who underwent liver MRI, hepatic steatosis was identified by a proton density fat fraction of ≥5%, as per AASLD criteria. Among patients diagnosed with steatotic liver disease, 7.9% were classified as having MetALD, while 2.2% were diagnosed with ALD. Notably, the prevalence of MetALD was found to be similar in both obese men and women, whereas it was absent in nonobese women [33].

Currently, data regarding the specific impact of various cardiometabolic risk factors and alcohol consumption on the development of liver steatosis, fibrosis, cirrhosis, and hepatocellular carcinoma are limited. This underscores the importance of ongoing research efforts to elucidate the complex interactions between metabolic dysfunction, alcohol consumption, and liver disease progression in individuals with MetALD [34].

### 2.4. Genetics

The heritability of hepatic steatosis has been evaluated in a prospective twin study, which showed data of 0.52 (95% CI 0.31–0.73; *p* < 1.1 × 10^−11^) for steatosis and 0.5 (95% CI 0.28–0.72; *p* < 6, 1 × 10^−11^) for liver fibrosis [35]. Single nucleotide polymorphisms (SNPs) in MASLD have been studied extensively, where several loci have been associated with susceptibility and progression [36].

### 2.5. PNPLA3 (Patatin-Like Phospholipase Domain Containing 3)

Multiple genetic factors have been found to increase steatohepatitis risk; the best characterized is the SNP in the patatin-like phospholipase domain containing 3 (PNPLA3) gene [37], also known as adiponutrin or calcium-independent phospholipase A2-epsilon, encodes the amino acid substitution I148M and regulates lipolysis in hepatocytes. The risk associated with the PNAPLA3-I148M variant is resistance to expected proteasomal degradation and lipid accumulation [37].

PNPLA3 is a multifunctional enzyme with TAG lipase and acylglycerol O-acyltransferase activity, and the rs738409 variant is associated with this function loss [36]. It has been described that PNPLA3 is necessary for hepatic stellate cell activation and that the 148M genetic variant enhances its profibrogenic characteristics, providing a greater risk of disease progression in carriers [38].

The PNPLA3 I148M variant, induced by a cytosine to guanine nucleotide transversion mutation (SNP rs738409), was associated with hepatic increased fat content and inflammation. This mutation was found more frequently in Hispanics (49%), in addition to having an association with alanine aminotransferase elevation [39]. On the other hand, the rs738409 GG genotype is associated with steatosis grade > 1 (OR 1.35, 95% CI 1.04–1.76), NASH (OR 1.5, 95% CI 1.12–2.04) and fibrosis stage >1 (OR 1.5, 95% CI 1.09–2.12), but the 148M allele was associated with ALT levels in a dose-dependent manner [40].

Carrying the minor allele (G) rs738409 PNPLA3 mutation conferred an additive risk of hepatocellular carcinoma (HCC) (OR 2.26 [95% CI 1.23–4. 14], *p* = 0.0082). The risk of HCC conferred by the PNPLA3 genotype is not solely mediated through progression to advanced fibrosis [41]. Also, adiposity influences the effect of the M variant on hepatic TAG content and serum ALT [42].

### 2.6. TM6SF2 (Transmembrane Superfamily 6-Member 2)

The transmembrane superfamily 6-member 2 (TM6SF2) gene is required to mobilize neutral lipids for the assembly of VLDL (very low-density lipoproteins). The C allele (Glu167) is associated with increased cardiovascular risk by increasing circulating LDL cholesterol; on the other hand, the T allele (Lys167) is associated with MASLD [43]. TM6SF2 siRNA inhibition is associated with circulating TAG reduction and increased cellular TAG concentration, while TM6SF2 overexpression reduces liver cell steatosis [44].

The E167K substitution is a loss-of-function variant that results in impaired hepatic TAG secretion and hepatic fat accumulation. A study conducted with patients carrying the TM6SF2 E167K variant showed fatty liver because of reduced VLDL secretion. These patients were shown to have more severe steatosis and be more likely to have steatohepatitis (OR: 1.84; 95% CI: 1.23–2.79) and advanced fibrosis (OR, 2.08; 95% CI: 1.20). −3.55 (116). Adiposity has also been found to have an amplifying effect on the TM6SF2 variant [42].

### 2.7. GCKR (Glucokinase Regulatory Protein)

The glucokinase regulatory protein (GCKR) is a negative regulator of glucokinase; the P446L variant is related to loss of function, resulting in increased glucose phosphorylation and fatty acid synthesis in the liver [42]. Evidence shows that having this variant has a ~1.2 times increased risk of developing MASLD [36]. One study found that GCKR rs780094 is significantly associated with a MASLD increased risk (OR 1.25, 95% CI 1.14–1.36) [45]. Another study found that GCKR rs780094 is associated with higher MASLD scores and higher serum TAG [46]. As has been observed for the PNPLA3 and TM6SF2 variants, the effect of the GCKR risk allele is significantly amplified with increasing body mass index [42].

### 2.8. MBOAT7 (Membrane-Bound O-Acyltransferase Domain-Containing 7)

The membrane-bound O-acyltransferase domain-containing 7 (MBOAT7) gene is an integral membrane protein that increases phospholipid desaturation through free arachidonic acid [47] and predisposes to cirrhosis in patients with alcohol abuse. The genotype rs641738 in MBOAT7 is associated with increased hepatic fat content, more severe liver damage, and increased risk of fibrosis [48]. A study conducted in Italy found that the rs641738 T allele is associated with hepatocellular carcinoma in MASLD (OR 1.65, 95% CI 1.08–2.55), particularly in patients without advanced fibrosis [49].

### 2.9. HSD17B13 (Hydroxysteroid 17-Beta Dehydrogenase 13)

*HSD17B13* encodes hydroxysteroid 17-β dehydrogenase 13, a protein involved in the regulation of lipid biosynthetic processes and that has enzymatic activity for several bioactive lipid species implicated in lipid-mediated inflammation [50]. HSD17B13 has gained attention in recent years due to its association with liver disease, particularly MASLD and its more severe form, steatohepatitis. Research suggests that genetic variants of HSD17B13 may influence liver fat accumulation and the progression of liver disease. Specifically, certain variants have been associated with a reduced risk of MASLD and steatohepatitis, while others may confer an increased risk. However, the exact mechanisms by which HSD17B13 variants impact liver disease are still under investigation. Some studies suggest that HSD17B13 may play a role in lipid metabolism and inflammation in the liver [51].

### 2.10. PGC1α (Peroxisome Proliferator-Activated Receptor Gamma Coactivator 1-Alpha)

PGC-1s are essential coordinators of many vital cellular events, including mitochondrial functions, oxidative stress, endoplasmic reticulum homeostasis, and inflammation, and are key regulators of cellular energy metabolism and mitochondrial function. They play a crucial role in various metabolic processes, including fatty acid oxidation, gluconeogenesis, and mitochondrial biogenesis. Studies have shown that PGC1α expression is dysregulated in individuals with metabolic disorders such as obesity, insulin resistance, and MASLD. In the context of MASLD, alterations in PGC1α activity or expression may contribute to impaired mitochondrial function, decreased fatty acid oxidation, and increased hepatic lipid accumulation, thereby exacerbating liver steatosis and metabolic dysfunction [52].

### 2.11. SIRT1 (Sirtuin 1)

SIRT1 is a NAD+-dependent protein deacetylase that regulates various cellular processes, including metabolism, oxidative stress, and inflammation. Emerging evidence suggests that SIRT1 plays a critical role in the pathogenesis of metabolic disorders and liver diseases, including MASLD/steatohepatitis. SIRT1 activity is known to be reduced in individuals with obesity, insulin resistance, and liver steatosis. This dysregulation may contribute to impaired hepatic lipid metabolism, increased oxidative stress, and inflammation, ultimately promoting the progression of MASLD [53] (Table 1).

### 2.12. Epigenetics

Epigenetics comprises the molecular mechanisms that govern genetic expression in response to environmental stimuli, orchestrating phenotypic changes without altering the DNA sequence and persisting through cell division [47,54]. In the context of MASLD development, epigenetic nutritional mechanisms, particularly histone modifications such as methylation, acetylation, and phosphorylation, have garnered significant attention [55].

DNA methylation, a pivotal epigenetic modification, involves the addition of a methyl group at cytosine and guanine sites, mediated by methyltransferase enzymes utilizing bioactive nutrients like folate, methionine, serine, betaine, choline, retinoic acid, resveratrol, turmeric, sulforaphane, and polyphenols [56]. This process regulates gene expression by modulating the balance between S-adenosyl L-methionine (a methyl donor) and S-adenosyl homocysteine (an inhibitor of transmethylation reactions), thus influencing various aspects of glucose metabolism, lipid regulation, and oxidative stress [54].

Folate deficiency, for instance, induces hepatic steatosis by impairing genes involved in fatty acid biosynthesis, thereby increasing de novo lipogenesis and reactive oxygen species production. In animal models, methyl-deficient diets have been shown to predispose to hepatic steatosis through impaired phosphatidylcholine synthesis [54,57,58]. Moreover, epigenetic changes resulting from hypercaloric or low-methyl diets have been implicated in MASLD progression [57].

Hypercaloric diets produce epigenetic changes in mouse models through changes in DNA methylation that predispose to MASLD and progression to fibrosis; on the other hand, low-methyl diets contribute to the development of cirrhosis and liver cancer in rodents [59]. In a study with biopsies from patients with MASLD, the cytosine methylation status of the NADH dehydrogenase gene was evaluated, finding that hypermethylation correlated inversely with the level of physical activity [60].

These epigenetic patterns not only serve as potential diagnostic biomarkers but also offer promising therapeutic targets, given the reversibility of metabolic outcomes affecting the epigenome. Bioactive food components exert influence on gene expression by acting as sources of methyl groups or coenzymes in carbon metabolism, thus modulating histone modification enzymes. Understanding and targeting these epigenetic mechanisms hold great potential in managing MASLD and other metabolic disorders.

### 2.13. The Diet as a Triggering Factor for MASLD

The increased MASLD prevalence is strongly related to overnutrition and nutrient imbalance, in parallel to the increase in obesity and components of metabolic syndrome; for this reason, diet is a key risk factor in MASLD development [61]. Several studies discuss the relationship between different diets and MASLD, reporting that the Western diet with an excess of simple carbohydrates and saturated fats is the leading cause of obesity and liver disorders, such as hepatic steatosis. On the other hand, in most of the studies reviewed in patients with MASLD, fiber and antioxidant intake decreased [62]. Thus, individuals with obesity had more repercussions in the increase in de novo lipogenesis and the association with liver damage based on the Western dietary pattern as a potential pro-inflammatory agent [63].

Frequent food consumption has been identified in patients with MASLD, such as excessive fast-food intake, red meat, sweets, whole dairy products, refined cereals, and sugary drinks, and a decrease in the group of vegetables, fruits, legumes, and whole grains [63]. The literature reports an association between the intake of simple carbohydrates and highlights an association between saturated fats and cholesterol intake with a decrease in polyunsaturated fatty acids [62].

Furthermore, consuming sugar-sweetened beverages was strongly associated with increased hepatic steatosis because fructose, high-fructose corn syrup, sucrose, and starch syrup elevated hepatic TAG synthesis by increasing fructose kinase expression and fatty acid synthase in patients with MASLD [64]. Fructose can induce de novo lipogenesis by the upregulation of sterol regulatory element-binding protein-1c and elemental carbohydrate response that regulate several lipogenic genes [65]. Therefore, food with a high glycemic index increases de novo lipogenesis, hypertriglyceridemia, insulin, and hepatic steatosis [66].

Consumption of saturated fat derived from red meat exceeds approximately five times the recommended daily intake in patients with MASLD, which increases conjugated linoleic acid in liver cells and promotes stress on the endoplasmic reticulum [67]. On the other hand, there is a theory that excessive consumption of heme iron derived from red meat may play a role in MASLD pathogenesis by increasing oxidative stress [62]. Excess sodium consumption, derived from greater consumption of meats, fats, and ultra-processed products, has been detected [66].

The effect of a Western diet can cause MASLD through various metabolic pathways. First, alterations of adipokines in adipose tissue, with increased leptin, resistin, and hypo adiponectin associated with high-carbohydrate diets and fatty foods, lead to increased lipogenesis, gluconeogenesis, and free fatty acid absorption in the liver and adipose tissue [17]. Other effects on the pancreas are a decrease in insulin sensitivity and an increase in lipogenesis and gluconeogenesis, leading to inflammation mediated by Kupffer cells, C-reactive protein, and tumor necrosis factor α, IL-1b, IL-4, IL-6, IL-10 related to steatohepatitis [63] (Figure 1).

## 3. Dietary Treatments

Lifestyle modification is supported as a primary therapy to control MASLD and steatohepatitis as obesity is closely related to this spectrum of liver diseases; therefore, weight loss is considered essential [68,69]. The clinical practice guidelines propose recommendations for the treatment of patients with MASLD, identifying various dietary patterns that include calorie restriction, manipulation of macronutrients, and the exclusion of ultraprocessed foods and drinks with high fructose content [68]. A weight loss of >5% has been shown to reduce steatosis, a loss of >7% improves steatohepatitis histology, and a weight loss of >10% may impair fibrosis regression [70] (Table 2).

### 3.1. Caloric Restriction

Dietary energy is essential to hepatic steatosis; several studies have shown that calorie-restricted diets improve MASLD [69]. There is evidence from randomized studies in patients with obesity that compare low-carbohydrate or low-fat diets, which reduce hepatic steatosis [71]. The gradual reduction achieved through calorie restriction, regardless of the diet macronutrient composition, reaches a 7–10% weight loss goal that leads to a significant reduction in liver fat, resulting in an improvement in liver enzymes and histology [68].

There are different types of calorie restriction; most appear to be effective in long-term weight loss; however, macronutrient composition is reported to be even more important than total calories, demonstrating that changes in dietary composition alone can reduce fat infiltration. The controversy over hepatic fat infiltration and metabolic health changes differs when comparing the effect of a carbohydrate-restricted diet versus a fat-restricted diet [72].

A randomized clinical trial in adults found that a carbohydrate-restricted diet compared to a low-fat diet resulted in a similar weight loss but a more significant reduction in liver fat (−55% ± 14%, *p* < 0.001) [73]. In another study, patients receiving a low-carbohydrate diet had lower ALT concentrations than those receiving a high-carbohydrate or low-fat diet despite equal weight loss [6]. This suggests that a hypocaloric, low-carbohydrate diet may benefit MASLD patients, regardless of weight loss.

A 12-month randomized clinical trial included a low-fat diet combined with low-intensity exercise-induced weight loss and MASLD improvement, as well as resolution of steatohepatitis in 10% and liver fibrosis in 16% [74]. On the other hand, another 12-month randomized trial with a low-carbohydrate/low-fat diet along with moderate-intensity exercise produced a notable MASLD regression in subjects who lost 5–7% and a decrease in steatohepatitis in those who lost 7–10% [75].

Evidence indicates that carbohydrate-restricted diets may result in a more substantial reduction in liver fat compared to low-fat diets, even with similar weight loss. Furthermore, hypocaloric, low-carbohydrate diets may benefit MASLD patients independent of weight loss, as indicated by improvements in liver enzyme concentrations. Combining dietary interventions with exercise can further enhance the effectiveness of MASLD management.

### 3.2. Intermittent Fasting

Intermittent fasting, also known as time-restricted eating, consists of allowing food consumption in a specific time window during the day followed by a fasting window. The most popular intermittent fasting program is 16:8, in which one fasts for 16 h and eats freely in the remaining 8 h of the day; however, there are other strategies, such as 5:2 fasting, where for two nonconsecutive days, only 25% of the daily kcal is ingested per day and on the other days an ad libitum diet is followed [76].

Both strategies have shown benefits for reducing intrahepatic fat and a long list of biochemical lipid profile parameters and liver function tests [76]. The explanation for the advantages of intermittent fasting is the use of glucose, derived from glycogenolysis, and the metabolic change that occurs between 12–36 h of fasting for the conversion to ketones derived from fatty acids [77].

Alternate-day fasting is a new dietary therapy that shows intrahepatic fat decrease and improves serum parameters, favoring patients with MASLD. On the other hand, mice fasting for 24/24 h are positive in decreasing hepatic lipogenesis and increasing b-oxidation markers, hepatic steatosis, and inflammation reduction [78].

Intermittent fasting, a technique and not a dietary intervention, can be combined with caloric restriction to obtain better results, achieving a more significant decrease in total body fat mass and a rapid improvement in intrahepatic fat [77]. With this information, it could be said that intermittent fasting can be an additional factor in treating metabolic problems and hepatic steatosis. Intermittent fasting has been gaining momentum lately after years of being widely criticized for posing a health risk. The physiological principle of intermittent fasting consists of a metabolic switch towards the preferential utilization of fatty acids, resulting in fat mobilization [79]. It can be an alternative to continuous caloric restriction, with similar weight loss [80]. Intermittent fasting not only aids in weight loss but also offers additional benefits by improving metabolic syndrome. It achieves this by redistributing fat, particularly reducing abdominal fat, and lowering key cardiovascular risk factors such as blood pressure, heart rate, cholesterol, and triglyceride levels. These improvements are largely attributed to the resultant weight loss [81].

In conclusion, intermittent fasting has emerged as a promising dietary strategy for improving metabolic health and reducing hepatic steatosis. By structuring eating patterns to include defined fasting periods, intermittent fasting induces metabolic changes that promote the utilization of fatty acids for energy, leading to fat mobilization and subsequent reductions in intrahepatic fat. Research suggests that intermittent fasting not only improves liver function and lipid profiles but also contributes to overall weight loss and metabolic syndrome management. Further research is needed to elucidate the long-term effects and optimal implementation of intermittent fasting for the management of metabolic-associated fatty liver disease and related metabolic conditions. Nonetheless, the accumulating evidence suggests that intermittent fasting holds promise as a valuable tool in the multifaceted approach to improving metabolic health and combating hepatic steatosis.

### 3.3. Ketogenic Diet

The ketogenic diet is a low-carbohydrate, high-fat diet that emerged in the 1920s and has been used as a therapeutic approach for various disorders [82]. It has been proposed as an effective intervention for weight loss and its efficacy in metabolic syndrome, including potentially suitable for MASLD treatment [83].

The European Food Safety Authority stated that the ketogenic diet consists of a restricted caloric intake with an amount of protein of 1.2–1.4 g/kg of about 75 g/day, highlighting a drastic carbohydrate reduction of <30–50 g/day and fat limitation of approximately 20 g/day [84]. The ketogenic diet is an intervention that develops ketosis from glycogenolysis, leading to a greater reduction in intrahepatic lipids as they rapidly deplete liver glycogen [85]. Ketone bodies have been proposed as valuable modulators of inflammation and fibrosis, although there is still controversy regarding their benefits in MASLD [84].

The hypocaloric ketogenic diet is related to increased adiponectin levels after eight weeks. Adiponectin exerts an anti-inflammatory effect and modulates insulin sensitivity by stimulating the use of glucose and the oxidation of fatty acids; it also reduces the levels of TNF-α, glycosylated hemoglobin (HbA1c), lipid profile (triglycerides, total cholesterol, and LDL) and with increased levels of low-density lipoprotein (HDL) and IL-10 (inflammatory mediator) [86].

Clinical trials lasting two weeks emphasize that the ketogenic diet reduced liver fat content more than a standard hypocaloric diet [74]. A six-month pilot study with the ketogenic diet evaluated its effects on obesity-associated MASLD, showing improvements in steatosis, necroinflammatory degree, and fibrosis, corroborated by biopsy [87].

Whether a ketogenic diet decreases MASLD in mice has not yet been determined. Although mice lose weight and develop ketosis, metabolic effects have been evaluated, having an unfavorable energy balance and genetic expression [88]. Garbow et al. determined a correlation between the ketogenic diet in mice for 12 weeks, intrahepatic fat, and MASLD. Mice fed the ketogenic diet were lean, euglycemic, ketotic, and hypoinsulinemic but were glucose intolerant and exhibited MASLD due to ER stress, cellular injury, and macrophage accumulation [89].

The recommendations for the implementation of a ketogenic diet are plant-origin protein increase and animal-origin protein decrease, implementation of fermented foods and drinks (e.g., yogurt, kefir, kimchi), probiotics and omega 3 fatty acids, including unsaturated fatty acids in adequate quantity and quality and avoiding sweeteners (except stevioside) and processed foods [90]. Some of the absolute contraindications are type 1 diabetes mellitus, latent autoimmune diabetes in adults, B cell insufficiency in type 2 diabetes mellitus, use of sodium/glucose cotransporter 2 inhibitors (due to the risk of euglycemic diabetic ketoacidosis), pregnancy, breastfeeding, renal, hepatic, respiratory and cardiac failure (NYHA II-IV), etc. [91].

In conclusion, the ketogenic diet has garnered attention as a therapeutic approach for various conditions, including MASLD. By inducing ketosis through drastic carbohydrate restriction and moderate protein intake, the ketogenic diet promotes the utilization of fatty acids for energy, leading to reductions in intrahepatic lipids. However, further research is needed to elucidate its long-term efficacy, safety, and optimal implementation strategies. Individualized recommendations and close monitoring are essential to ensure that potential benefits outweigh the risks for each patient.

### 3.4. Mediterranean Diet

The Mediterranean diet is a nutritional model that originates in the Mediterranean Sea’s coastal states. It consists of reducing the consumption of saturated fats and animal proteins, increasing the contribution of antioxidants, fiber, and monounsaturated fatty acids, in addition to an adequate balance of omega-3/omega-6, which makes it rich in polyunsaturated fats, polyphenols, vitamins, and carotenoids, with their anti-inflammatory and antioxidant effects [92,93].

Several studies have analyzed the beneficial effects of different diets on MASLD, mainly the Mediterranean Diet, producing a significant improvement in steatosis, even in the absence of weight loss [94,95]. The EASL-EASD-EASO clinical practice guidelines recommend the Mediterranean diet as the ideal non-pharmacological intervention and lifestyle option for MASLD treatment [93].

Dr Kontogianni’s research group was the first to explore the Mediterranean diet and NAFLD, finding that greater adherence to the Mediterranean diet correlated with lower insulin resistance, serum alanine aminotransferase concentrations, and MASLD severity [96]. Evidence from epidemiological studies indicates that greater adherence to the Mediterranean diet is associated with a lower likelihood of having metabolic syndrome or progressing to steatohepatitis [97].

A study with overweight/obese diabetic patients treated with a Mediterranean diet showed a decrease from 93% to 48% in hepatic steatosis, in addition to reducing metabolic parameters and liver enzymes [98]. A randomized controlled clinical trial confirms that Mediterranean lifestyle changes and weight loss may be a treatment option for patients with MASLD [97].

In conclusion, the Mediterranean diet stands out as a highly beneficial nutritional model for individuals with MASLD. By emphasizing the consumption of antioxidants, fiber, monounsaturated fatty acids, and polyunsaturated fats, while reducing saturated fats and animal proteins, this diet exerts potent anti-inflammatory and antioxidant effects, which are particularly advantageous for managing MASLD, offering not only hepatic benefits but also contributing to overall health and well-being.

### 3.5. Paleolithic Diet

This combination results in a diet rich in foods with antioxidant, antifibrotic, and immunomodulatory properties, such as fiber, potassium, vitamins B, D, E, and K, polyphenols, monounsaturated and polyunsaturated fatty acids, which are involved in MASLD prevention [94,99]. Furthermore, adherence to this diet, even in the short term, helps normalize blood pressure, improve lipid profile (reduction in total cholesterol, triglycerides, low-density lipoproteins, LDL), increase insulin sensitivity, decrease weight, fat mass, and waist circumference [100].

The Paleolithic diet is similar to what hunter-gatherer ancestors consumed; it is characterized by a high intake of vegetables, fruits, oilseeds, eggs, fish, and lean meats and excludes processed and ultra-processed products. This combination results in a diet rich in foods with antioxidant, antifibrotic, and immunomodulatory properties, such as fiber, potassium, vitamins B, D, E, and K, polyphenols, monounsaturated and polyunsaturated fatty acids, which are involved in the prevention of MASLD [94,99]. Short-term adherence to this diet improves glucose levels, insulin sensitivity, blood pressure, and lipid profile with or without weight loss [100,101].

A two-year study with the Paleolithic diet, compared to a conventional low-fat diet, showed favorable effects for the Paleolithic diet in reducing hepatic steatosis and insulin sensitization in the liver [102].

The Paleolithic diet, modeled after the dietary habits of our hunter-gatherer ancestors, presents a compelling nutritional approach for the prevention and management of MASLD. By emphasizing whole, unprocessed foods rich in antioxidants, fiber, vitamins, and healthy fats while excluding processed and ultra-processed products, the Paleolithic diet provides a diverse array of nutrients with antioxidant, antifibrotic, and immunomodulatory properties.

### 3.6. High Fiber Diet

Diets with adequate fiber intake and caloric restriction are linked to weight loss and MASLD regression, as they can reduce appetite and meal frequency by downregulating the hormone ghrelin, which controls hunger and body weight. Furthermore, high-fiber diets reduce prostaglandins E2 and modulate the expression of sterol regulatory element binding protein 1 (SREBP1), which plays a vital role in developing metabolic diseases such as MASLD [103]. The type of fiber that has the most evidence to improve insulin sensitivity is soluble fibers, which are found in nuts, seeds, beans, lentils, peas, etc. [104].

A prospective study of 115 patients, who were followed for 8.6 months, evaluated the comparison of a high-fiber diet and aerobic exercise versus exercise alone. The high-fiber diet was considered 12 g of fiber per day, and physical exercise was performed 2 to 3 times a week for 30 to 60 min. It was concluded that a high-fiber diet and exercise effectively reduce intrahepatic fat [105].

Diets rich in fiber, coupled with caloric restriction and exercise, represent promising strategies for weight loss and regression of MASLD. Adequate fiber intake has been associated with reduced appetite and meal frequency, partly through the downregulation of ghrelin, a hormone that controls hunger and body weight. Further research is warranted to explore the long-term effects and optimal strategies for implementing fiber-rich diets in MASLD populations. Nonetheless, the evidence suggests that dietary fiber intake, along with lifestyle modifications, holds promise as a valuable component of multifaceted interventions for MASLD.

### 3.7. High Protein Diet

High protein diets can be considered as those in which the distribution of macronutrients is 30% protein, 40% carbohydrates, and 30% fat. High protein, isocaloric, and hypercaloric diets effectively reduce intrahepatic fat and improve lipid profile, glucose homeostasis, and liver enzymes in patients with MASLD, regardless of the weight loss and initial weight [106,107]. The beneficial effects of this type of intervention have been related to lipolysis downregulation, increased mitochondrial activity, activation of fatty acids b-oxidation, increased insulin sensitivity, and decreased fibroblast growth factor 21 [108]. 

A high-protein diet, primarily sourced from vegetables and low in carbohydrates and sugar, shows promise as a therapeutic strategy to reverse the phenotype of MASLD and alleviate insulin resistance. This approach exhibits potential efficacy, contingent upon the preservation of liver function and muscle catabolism [109].

In conclusion, high-protein diets with a macronutrient distribution of approximately 30% protein, 40% carbohydrates, and 30% fat have demonstrated significant benefits for patients with MASLD. Both isocaloric and hypercaloric versions of these diets have shown effectiveness in reducing intrahepatic fat, improving lipid profiles, glucose homeostasis, and liver enzymes, irrespective of weight loss and initial weight. Incorporating adequate protein into the diet may represent a promising strategy in the comprehensive management of MASLD.

## 4. Other Treatments

Lifestyle, diet, and exercise interventions are currently recommended to manage MASLD. Despite its epidemic proportions, no approved treatment to treat MASLD is currently available. This section will focus on describing current interventions for the treatment of MASLD (Table 2).

### 4.1. Exercise

Regardless of weight loss, physical exercise has beneficial effects at the liver and cardiometabolic levels and is currently considered first-line therapy for MASLD [110]. Current recommendations are to perform moderate exercise 3 to 5 times per week for a total of 150 min to prevent or improve MASLD; in steatohepatitis, it is recommended to perform physical activity with greater intensity to stop or reverse the disease; however, it is necessary to adapt this intervention to the preferences of each patient [95].

Some of the mechanisms by which exercise shows benefits in MASLD are that it increases fatty acids β-oxidation, induces autophagy, overexpresses peroxisome proliferator-activated receptor gamma (PPAR-γ), which has a potent effect on cellular sensitization to insulin, prevents mitochondrial damage and attenuates hepatocyte apoptosis [111,112].

Dietary intervention with exercise is the best-known strategy for weight loss and MASLD treatment, approved by the American Association for the Study of Liver Diseases and the American Gastroenterological Association [110]. A randomized control trial in patients with steatohepatitis reported that intervention with moderate-intensity exercise and a reduced-calorie diet reduced steatohepatitis activity score on liver biopsy [113]. Also, significant regression in liver fibrosis has been informed, demonstrated by liver biopsies, after an aerobic exercise intervention for 12 weeks [114]. A meta-analysis of clinical trials suggests that physical activity with dietary treatment improves liver steatosis, liver enzymes, glucose, and serum lipid metabolism [115].

The best available evidence is that in patients with MASLD, physical exercise can modify the de novo lipid synthesis, stimulating the phosphorylation of several enzymes involved in the conversion of acetyl-CoA into free fatty acids, which inhibit their expression. On the other hand, epigenetic mechanisms (reduced DNA hypermethylation) have been proposed to be responsible for the effect of physical exercise on metabolic pathways, including de novo lipogenesis [111].

Different doses and intensities of physical activity modify hepatic steatosis; even the group with the lowest dose and intensity shows a reduction in visceral and hepatic fat regardless of weight loss [116]. High-intensity exercise modifies intrahepatic triglycerides and visceral lipids in patients with MASLD, improving energy expenditure, fatty acid oxidation, and reducing visceral adipose fat [117].

In conclusion, physical exercise stands as a cornerstone in the management of MASLD, showcasing beneficial effects at both hepatic and cardiometabolic levels. Further research is warranted to optimize exercise prescriptions and elucidate underlying mechanisms, but current evidence strongly supports its integral role in MASLD treatment strategies.

### 4.2. Bariatric Surgery

Bariatric surgery is an obesity treatment that causes weight loss with possible reductions in metabolic syndrome, in addition to reducing hepatic fat, inflammation, and fibrosis [118]. The mean percentage of excess weight loss with different types of bariatric surgery is 61.2% (95% CI: 58.1–64.4%) [119]. This surgery is indicated in people with morbid obesity (BMI ≥ 40 kg/m^2^) or a BMI between 35 and 40 mg/m^2^ with comorbidities associated with metabolic syndrome [120].

A prospective study found that one year after bariatric surgery, body mass index was significantly decreased, and steatohepatitis disappeared in 85% of the population (95% CI, 75.8–92.2%). Another retrospective study evaluated the resolution of steatohepatitis five years after bariatric surgery and found that 84% (95% CI: 73.1–92.2%) resolved steatohepatitis without worsening fibrosis, and fibrosis decreased by 70.2% (95% CI: 56.6–81.6%) [121].

Bariatric surgery represents a highly effective treatment option for individuals with MASLD, particularly those with morbid obesity or a high body mass index coupled with comorbidities associated with metabolic syndrome. This surgical intervention not only induces significant weight loss but also leads to reductions in hepatic fat, inflammation, and fibrosis.

### 4.3. Pharmacological Treatment and Dietary Supplements

Dietary changes and lifestyle modifications are first-line therapy; however, many patients cannot reduce body weight. Therefore, the question is which population should be offered pharmacological treatment for NAFLD/NASH. According to European practice guidelines and AASLD guidance, pharmacotherapies should be considered for NASH patients with stage 2 or higher fibrosis and early-stage fibrosis with a high risk of fibrosis progression [132].

Randomized controlled trials have been designed to evaluate improvement in liver inflammation or fibrosis as a primary outcome; however, no specific pharmacological treatment can be recommended now [68,133].

### 4.4. Pioglitazone

Drugs targeting PPAR, such as pioglitazone, are used clinically to treat type 2 diabetes. In the PIVEN Strial study, treatment of nondiabetic subjects with NASH with pioglitazone (30 mg/day) reduced hepatic steatosis, lobular inflammation, hepatocellular distention, improved insulin resistance, and liver enzyme levels; however, it was associated with more significant body weight gain. On the other hand, pioglitazone is contraindicated in patients with established heart failure or at increased risk of heart failure [122].

The 2023 AASLD practice guideline states that pioglitazone can treat patients with and without type 2 diabetes with biopsy-proven MASLD. Pioglitazone improves NASH and can be considered for patients with NASH in the context of patients with T2DM [95].

Low doses of pioglitazone (15 mg/day) are being evaluated in a phase II clinical trial (Clinicaltrials.gov, NCT04501406) to evaluate the effect of pioglitazone on liver histology in patients with NASH. Clinical trials evaluating other PPAR agonists are underway [123].

### 4.5. Liraglutide

Liraglutide has been the most studied GLP-1RA drug in MASLD, which can improve IR and reduce liver fat content in patients with T2DM [124]. In obese diabetic rats, liraglutide can enhance PPARα expression through a GLP-1/AMPK receptor signaling pathway [125]. A phase II clinical trial showed that subcutaneous injection of liraglutide at a dose of 1.8 mg/d in patients with NASH manifested regression of MASLD without progression of liver fibrosis. Liraglutide had a good safety profile and significantly improved liver function and histological features in NASH patients with glucose intolerance. However, the use of liraglutide is often associated with some mild gastrointestinal adverse reactions, such as diarrhea or constipation and loss of appetite [126]. A human study showed that treatment with GLP-1 agonists did not decrease the risk of developing NAFLD compared to treatment with insulin [127]. Therefore, more studies are needed to clarify the therapeutic drug of GLP-1 in NAFLD.

### 4.6. Resmetirom

In a phase 3 trial, resmetirom 80 mg, resmetirom 100 mg, and a placebo were administered, enrolling adults with biopsy-confirmed NASH and a fibrosis stage of F1, F2, or F3. Resolution of NASH without worsening fibrosis was achieved in 25.9% of patients in the resmetirom 80 mg group and 29.9% of those in the resmetirom 100 mg group, compared with 9.9%. 7% of those in the placebo group. Improvement of fibrosis in at least one stage without worsening of the NAFLD activity score was achieved in 24.2% of patients in the resmetirom 80 mg group and in 25.9% of those in the resmetirom group of 100 mg, compared with 14.2% of those in the placebo group. The change in low-density lipoprotein cholesterol levels from baseline to week 24 was −13.6% in the resmetirom 80 mg group and −16.3% in the resmetirom 100 mg group, compared with 0.1% in the placebo group (*p* < 0.001 for both placebo comparisons). Diarrhea and nausea were more common with resmetirom than with the placebo. Both resmetirom 80 mg and 100 mg doses were superior to the placebo with respect to the resolution of NASH and the improvement of liver fibrosis by at least one stage [128].

In conclusion, while dietary changes and lifestyle modifications remain the first-line therapy for MAFLD, there exists a subset of patients for whom achieving significant weight reduction may be challenging. For these individuals, pharmacological treatment becomes a consideration, particularly in cases of advanced fibrosis or a high risk of fibrosis progression. Overall, while pharmacological treatments for MAFLD are under investigation, further research is needed to establish their long-term efficacy and safety profiles. The decision to initiate pharmacotherapy should be made on a case-by-case basis, considering individual patient characteristics and treatment goals. Additionally, ongoing clinical trials evaluating other potential pharmacological agents may provide further insights into novel therapeutic approaches for MAFLD management.

### 4.7. Vitamin E

Vitamin E is a fat-soluble nutrient that acts as an antioxidant which helps protect cells against damage caused by free radicals. Supplementation with vitamin E 800 IU daily for 96 weeks has been shown to improve hepatic histology (≥2 points) assessed with the MASLD Activity Score (NAS) compared to the placebo in nondiabetic MASLD patients. The 30% decrease in aminotransferases compared to the initial value is related to improved histological parameters in patients supplemented with vitamin E. There is controversy over the possible risks of long-term vitamin E use, such as bleeding and hemorrhagic stroke and increased risk of prostate cancer, so its use should be discussed with patients before supplementation is initiated [95].

### 4.8. Omega 3

Omega-3 fatty acid supplementation effectively treats triglyceride-dominated dyslipidemias in patients with MASLD in combination with lifestyle changes; however, its use to reduce intrahepatic fat has no currently established benefits [95]. A double-blind, randomized, controlled clinical trial explored doses of 1728 mg of fish oil rich in polyunsaturated fatty acids (PUFA) such as EPA (eicosapentaenoic acid) and DHA (docosahexaenoic acid), which have been associated with multiple health benefits; however, did not show a reduction in intrahepatic fat measured by MRI in overweight men over 12 weeks [129].

### 4.9. Coffee

Coffee contains several chemical components, such as caffeine, diterphenocyclic alcohols, potassium, niacin, and chlorogenic acid, which have antioxidant and antifibrotic properties. A meta-analysis that included seven articles with a total of 4825 cases (coffee consumers) vs. 49,616 controls (noncoffee consumers) evaluated the consumption of 0 to >5 cups of coffee per day and its association with MASLD, demonstrating that consumption of more than three cups of coffee per day is related to a lower risk of developing MASLD, as well as fibrosis and cirrhosis [130]. Currently, international guidelines recommend consuming at least three cups of coffee a day (with or without caffeine) without contraindications [95]. The mechanism by which caffeine is proposed to prevent or reverse liver fibrosis is by acting as an antagonist of the adenosine A2A receptor that inhibits hepatic stellate cell activation [131].

Overall, while these dietary supplements hold potential as adjunctive therapies for MAFLD, further research is needed to understand their efficacy, safety, and optimal dosing regimens better. Clinicians should consider individual patient factors and preferences when discussing the use of these supplements as part of a comprehensive management approach for MAFLD.

## 5. Future Perspectives

The incidence of NAFLD and NASH is currently increasing, which is positively associated with the prevalence of obesity and diabetes, for which there are no approved pharmacological treatments currently approved.

To date, the only known effective treatment for both steatohepatitis and fibrosis regression is weight loss through diet and exercise or bariatric surgery. However, not all patients are candidates for bariatric surgery, and lifestyle changes have proven difficult to achieve, with reported adherence of <50% [134]. Pharmacological treatments, such as pancaspase inhibitors, CCR2/5 antagonists, PPAR, and FXR agonists, are potential targets for treating metabolic diseases, only serving as an adjuvant to lifestyle changes [135]. Future perspectives could be aimed at finding treatments that can reverse fibrosis, thus preventing cirrhosis, and are easy to follow.

On the other hand, early diagnosis of MASLD progression to liver fibrosis, cirrhosis, or HCC is of vital importance due to the difficulty of reversing advanced stages of the disease. Preclinical studies and clinical trials are underway to evaluate potential treatment options for NAFLD and NASH, with potential targets for treating metabolic diseases such as NAFLD and NASH. Combining treatments, such as combined medical treatment and physical activity, could reduce the treatment time and improve the outcome.

## 6. Discussion

MAFLD is primarily driven by obesity, with sedentary behavior and poor dietary habits exacerbating its progression. Reduced insulin sensitivity and heightened insulin secretion are common in individuals with steatohepatitis. Alcohol consumption also plays a significant role in liver disease pathogenesis.Genetic factors, including single nucleotide polymorphisms (SNPs) in genes related to lipid metabolism and inflammation, contribute to MAFLD susceptibility and progression.Epigenetic mechanisms, such as histone modifications and DNA methylation, further regulate gene expression and impact disease development.Dietary patterns high in simple carbohydrates, saturated fats, and sugar-sweetened beverages contribute to hepatic steatosis and inflammation, while inadequate intake of fiber-rich foods exacerbates metabolic dysregulation.Lifestyle modification, including weight loss through caloric restriction and manipulation of macronutrients, is essential in managing MASLD and steatohepatitis. Carbohydrate-restricted diets may offer benefits independent of weight loss.Additionally, dietary supplements and pharmacological treatments can be considered adjunctive therapies, particularly for patients who struggle with significant weight reduction.A multifaceted approach combining dietary interventions, lifestyle modifications, and, when necessary, pharmacological treatments, holds promise in addressing the complex pathophysiology of MAFLD and improving patient outcomes. However, further research is needed to establish their long-term efficacy and safety profiles.

## 7. Conclusions

The development and progression of MASLD are closely related to unhealthy lifestyle habits and are coupled with genetic and epigenetic mechanisms. Weight loss is considered the cornerstone of treatment, along with exercise, which increases the oxidation of fatty acids that promote hepatic steatosis. Therefore, lifestyle modification is considered the first-line therapy to control MASLD (Figure 2).

## Figures and Tables

**Figure 1 ijms-25-04397-f001:**
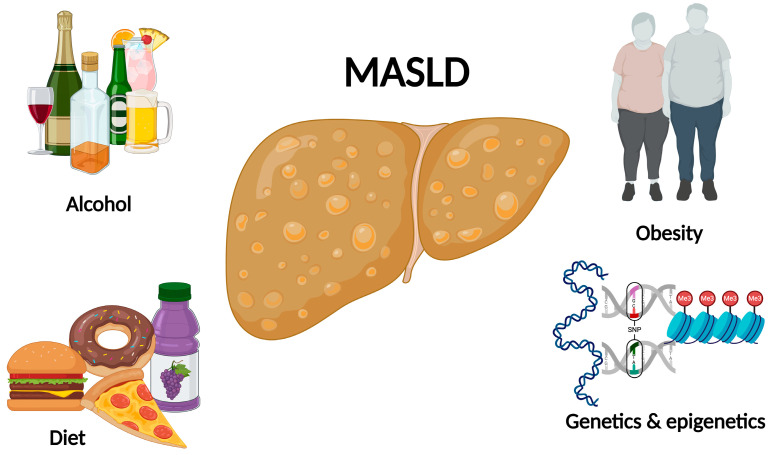
Risk factors for MASLD development.

**Figure 2 ijms-25-04397-f002:**
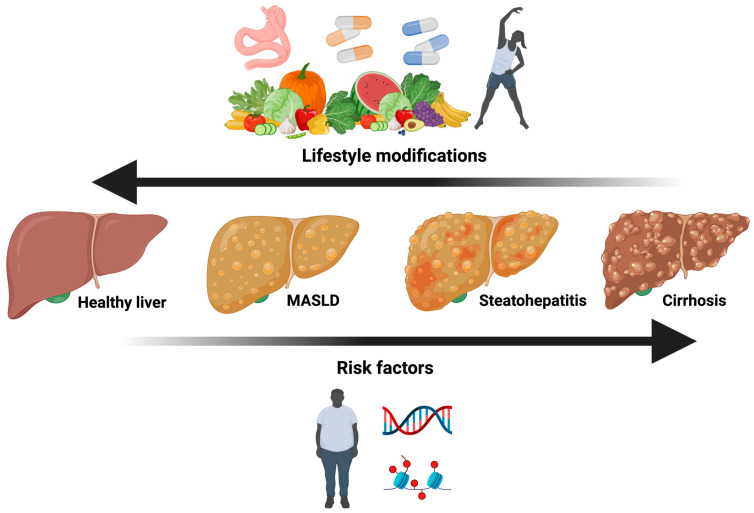
Treatment options for MASLD and the risk factors. The currently effective treatment options for MASLD and its progression include lifestyle modifications such as bariatric surgery, different types of diets, exercise, pharmacological treatments, and dietary supplements. Some of the main risk factors are a sedentary lifestyle and genetic and epigenetic factors.

**Table 1 ijms-25-04397-t001:** Genes involved in MASLD.

GENE	ACTION	MASLD RISK
PNPLA3	Regulates lipolysis in the hepatocyte lipid droplets. It is expressed in the liver and adipose tissue [38,39].	The associated risk to the PNAPLA3-I148M variant is resistance to expected proteasomal degradation and accumulation of lipid droplets [38].
TM6SF2	Localized in the liver and intestine, TM6SF2 protein is required to mobilize neutral lipids for VLDL (very low-density lipoproteins) assembly. TM6SF2 siRNA inhibition is associated with reduced TAG and increased cellular TAG concentration and lipid droplet content [44].	A study found that carriers of the TM6SF2 E167K variant were more likely to have steatohepatitis (OR: 1.84; 95% CI 1.23–2.79) and advanced fibrosis (OR, 2.08; 95% CI: 1.20–3.55) [43].
GCKR	Negative regulator of glucokinase; p.P446L is a loss-of-function variant that results in increased phosphorylation of glucose, glycolysis, and fatty acid synthesis in the liver [46].	A study found that the *GCKR* rs780094 is significantly associated with increased MASLD risk (OR 1.25, 95% CI 1.14–1.36) [45].
MBOAT7-TMC4	Integral membrane protein that increases phospholipid desaturation through free arachidonic acid [48].	Genotype rs641738 at the MBOAT7TMC4 is associated with increased hepatic fat content, more severe liver damage, and increased risk of fibrosis [49].
HSD17B13	A protein involved in regulating lipid biosynthetic processes, it has enzymatic activity for several bioactive lipid species implicated in lipid-mediated inflammation [50].	Liver fat accumulation and the progression of liver disease [51].
PGC1α	PGC-1s are essential coordinators of many vital cellular events, including mitochondrial functions, oxidative stress, endoplasmic reticulum homeostasis, and inflammation, which is a key regulator of cellular energy metabolism and mitochondrial function [52].	May contribute to impaired mitochondrial function, decreased fatty acid oxidation, and increased hepatic lipid accumulation [52].
SIRT1	NAD+-dependent protein deacetylase regulates various cellular processes, including metabolism, oxidative stress, and inflammation [53].	May contribute to impaired hepatic lipid metabolism, increased oxidative stress, and inflammation, ultimately promoting the progression of MASLD [53].

MASLD, Metabolic dysfunction-associated steatotic liver disease; PNPLA3, patatin-like phospholipase domain-containing 3; TM6SF2, transmembrane 6 superfamily member 2; GCKR, glucokinase regulatory protein; MBOAT7-TMC4, membrane-bound O-acyltransferase domain-containing 7-transmembrane channel-like 4; HSD17B13, hydroxysteroid 17-beta dehydrogenase 13; PGC1α, peroxisome proliferator-activated receptor gamma coactivator 1-alpha; SIRT1, sirtuin.

**Table 2 ijms-25-04397-t002:** Potential treatments for MASLD.

DIETARY TREATMENTS	BENEFITS
Caloric restriction	Improve MASLD, reduce hepatic steatosis, and improve liver enzymes [6,68,69,71,72,73,74,75].
Intermittent fasting	It reduces intrahepatic fat, improves metabolic syndrome, and reduces abdominal fat, blood pressure, heart rate, cholesterol, and triglycerides, favoring patients with MASLD [76,77,78,79,80,81].
Ketogenic diet	Weight loss, improvement in metabolic syndrome, steatosis, necroinflammatory degree and fibrosis [74,82,83,84,85,86,87,88,89,90,91].
Mediterranean diet	Significant improvement of steatosis in addition to reducing metabolic parameters and liver enzymes [92,93,94,95,96,97,98].
Paleolithic diet	Improves glucose levels, insulin sensitivity, blood pressure, and lipid profile and reduces hepatic steatosis [94,99,100,101,102].
High fiber diet	Reduces E2 prostaglandins, modulates the expression of SREBP1 and intrahepatic fat [103,104,105].
High protein diet	Reduce intrahepatic fat and improve lipid profile, glucose homeostasis, and liver enzymes [106,107,108,109].
OTHER TREATMENTS	BENEFITS
Exercise	Increases fatty acids β-oxidation, induces autophagy, overexpresses PPAR-γ, which has a potent effect on cellular sensitization to insulin, prevents mitochondrial damage, attenuates hepatocyte apoptosis, reduces visceral adipose fat, improves energy expenditure, liver steatosis, liver enzymes, glucose, and serum lipid metabolism [95,110,111,112,113,114,115,116,117].
Bariatric surgery	Reduces metabolic syndrome, hepatic fat, inflammation, and fibrosis [118,119,120,121].
Pharmacological treatments	
*Pioglitazone*	Reduced hepatic steatosis, lobular inflammation, hepatocellular distention, improved insulin resistance, and liver enzyme levels [95,122,123].
*Liraglutide*	Improvement of liver function and histological features and regression of MASLD without liver fibrosis progression [124,125,126,127].
*Resmetirom*	Improvement of fibrosis in at least one stage without worsening of NAFLD activity score and reduction in cholesterol and low-density lipoprotein levels [128].
Dietary supplements	
*Vitamin E*	Improvement of liver histology and decrease in aminotransferases [95].
*Omega 3*	Reduces dyslipidemia dominated by triglycerides [95,129].
*Coffee*	Prevent or reverse liver fibrosis [95,130,131].

MASLD, Metabolic Dysfunction-Associated Steatotic Liver Disease; SREBP1, sterol regulatory element binding protein 1; PPAR-γ, peroxisome proliferator-activated receptor gamma.

## Data Availability

No new data were created or generated in this manuscript. Data sharing is not applicable to this article.

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
