# Peer review of "The Relationship between Pathogenesis and Possible Treatments for the MASLD-Cirrhosis Spectrum"

_ijms, 2024, doi:10.3390/ijms25084397_

Round 1

Reviewer 1 Report

Comments and Suggestions for Authors

This manuscript by Vidal-Cevallos et al. provides a great review summarizing the development of Metabolic dysfunction-associated steatotic liver disease (MASLD) and the different forms of treatment that currently exist. I found the manuscript to be a great read, and very informative. Here are a few minor concerns:

1. The tables used to summarize information from the text are great, please also provide the references in Table 2 as done for Table 1

2. The manuscript would benefit from the addition of some illustrations/figures depicting the different pathways that are involved in the pathogenesis of the disease

3. Please add some bullets/numbering to the points being discussed. It will help make the discussion points easier to follow 

Author Response

This manuscript by Vidal-Cevallos et al. provides a great review summarizing the development of Metabolic dysfunction-associated steatotic liver disease (MASLD) and the different forms of treatment that currently exist. I found the manuscript to be a great read, and very informative. Here are a few minor concerns:

  1. The tables used to summarize information from the text are great, please also provide the references in Table 2 as done for Table 1

Answer (A): References corresponding to table 2 were added.

  1. The manuscript would benefit from the addition of some illustrations/figures depicting the different pathways that are involved in the pathogenesis of the disease.

A: A figure was made representing the risk factors for developing MASLD. (Figure 1)

  1. Please add some bullets/numbering to the points being discussed. It will help make the discussion points easier to follow.

A: A discussion section was added with the most relevant points of the manuscript.

Reviewer 2 Report

Comments and Suggestions for Authors

In the manuscript entitled “Relationship between pathogenesis and the possible treatments for the MASLD-Cirrhosis spectrum”, Paulina Vidal-Cevallos et al. reviewed the risk factors of MASLD and the mechanism, then they mentioned that diet is a triggering for MASLD and listed different kinds of diet and summarized the benefits of different diet. They also reviewed other treatments to manage MASLD, such as exercise and bariatric surgery. The topic is very meaningful.  

1.   The logic of risk factors part is a mess. The authors should make it more summarized and logic. The Genetics part, except the authors mentioned, there are more genes associated with MASLD, such as HSD17B13, PGC1α, SIRT1. The authors should review more. The Epigenetic part should be more summarized. The authors just put everything together, there is no logic.

2.   For the dietary treatment part, it’s a simple description of other researcher’s work, there is no summarization and conclusion.

Comments on the Quality of English Language

The authors should check the format and typo through the manuscript. For example, format in line 612 should be consistent with line 597, 627.

Author Response

In the manuscript entitled “Relationship between pathogenesis and the possible treatments for the MASLD-Cirrhosis spectrum”, Paulina Vidal-Cevallos et al. reviewed the risk factors of MASLD and the mechanism, then they mentioned that diet is a triggering for MASLD and listed different kinds of diet and summarized the benefits of different diet. They also reviewed other treatments to manage MASLD, such as exercise and bariatric surgery. The topic is very meaningful.  

  1. The logic of risk factors part is a mess. The authors should make it more summarized and logic. The Genetics part, except the authors mentioned, there are more genes associated with MASLD, such as HSD17B13, PGC1α, SIRT1. The authors should review more. The Epigenetic part should be more summarized. The authors just put everything together,there is no logic.

A: The entire risk factors section was edited. In the genetics part, the genes were separated and the genes HSD17B13, PGC1α, SIRT1 were added. The epigenetics section was edited and condensed for logic and understanding of the text.

  1. For the dietary treatment part, it’s a simple description of other researcher’s work, there is no summarization and conclusion.

A: In the dietary treatment part, conclusions were added in each section.

Comments on the Quality of English Language

The authors should check the format and typo through the manuscript. For example, format in line 612 should be consistent with line 597, 627.

A: A thorough review and correction of the entire manuscript was carried out.

Reviewer 3 Report

Comments and Suggestions for Authors

The paper entitled "Relationship between pathogenesis and the possible treatments for the MASLD-Cirrhosis spectrum" processes very important data in the management of MASLD. But I have a few comments:

1. The presentation of associated diseases is quite chaotic and needs to be more specific.

2. Alcohol as a risk factor is not included. It is already known that not only abuse but also low or moderate consumption poses a high risk. Please include this information.

3. Line 383: Intermittent fasting does not reduce abdominal fat, blood pressure, heart rate, cholesterol, and triglycerides. Weight loss has this consequence. Please clarify.

4. The ketogenic diet was used in intractable epilepsy in 1920 (doi: 10.3389/fnins.2019.00005), not in the treatment of metabolic syndrome. Please correct this.

5. High protein diets, particularly from vegetable sources, had the best effects (source: https://www.ncbi.nlm.nih.gov/pmc/articles/PMC6950466/). Please specify this in the appropriate section.

Author Response

The paper entitled "Relationship between pathogenesis and the possible treatments for the MASLD-Cirrhosis spectrum" processes very important data in the management of MASLD. But I have a few comments:

  1. The presentation of associated diseases is quite chaotic and needs to be more specific.

A: The entire section was restructured and improved.

  1. Alcohol as a risk factor is not included. It is already known that not only abuse but also low or moderate consumption poses a high risk. Please include this information.

A: In the risk factors section, a segment on alcohol use was added.

  1. Line 383: Intermittent fasting does not reduce abdominal fat, blood pressure, heart rate, cholesterol, and triglycerides. Weight loss has this consequence. Please clarify.

A: It was clarified that intermittent fasting does not reduce abdominal fat, blood pressure, heart rate, cholesterol and triglycerides, this reduction is due to weight loss.

  1. The ketogenic diet was used in intractable epilepsy in 1920 (doi: 10.3389/fnins.2019.00005), not in the treatment of metabolic syndrome. Please correct this.

A: The correction was made and the corresponding reference was added.

  1. High protein diets, particularly from vegetable sources, had the best effects (source: https://www.ncbi.nlm.nih.gov/pmc/articles/PMC6950466/). Please specify this in the appropriate section.

A: In the high protein diet section, it was added that vegetable protein is a good strategy to reverse MASLD with its corresponding reference.

Round 2

Reviewer 2 Report

Comments and Suggestions for Authors

In the manuscript entitled “Relationship between pathogenesis and the possible treatments for the MASLD-Cirrhosis spectrum”, Paulina Vidal-Cevallos et al. reviewed the risk factors of MASLD and the mechanism, and they mentioned and summarized how to improve MASLD. The topic is very meaningful.

The authors have edited the risk factors section and made conclusion. It’s good to publish.